# Does equivariance matter at scale?

**Johann Brehmer***  
*Qualcomm AI Research*†  
*mail@johannbrehmer.de*

**Sönke Behrends**  
*Qualcomm AI Research*†

**Pim de Haan***  
*Qualcomm AI Research*†

**Taco Cohen***  
*Qualcomm AI Research*†

**Reviewed on OpenReview:** *https://openreview.net/forum?id=wiLNute8Tn*

## Abstract

Given large datasets and sufficient compute, is it beneficial to design neural architectures for the structure and symmetries of each problem? Or is it more efficient to learn them from data? We study empirically how equivariant and non-equivariant networks scale with compute and training samples. Focusing on a benchmark problem of rigid-body interactions and on general-purpose transformer architectures, we perform a series of experiments, varying the model size, training steps, and dataset size. We find evidence for three conclusions. First, equivariance improves data efficiency, but training non-equivariant models with data augmentation can close this gap given sufficient epochs. Second, scaling with compute follows a power law, with equivariant models outperforming non-equivariant ones at each tested compute budget. Finally, the optimal allocation of a compute budget onto model size and training duration differs between equivariant and non-equivariant models.

## 1 Introduction

In a time of big data and abundant compute, how important are strong inductive biases? Consider problems governed by known symmetries: should one take these into account by designing and using equivariant neural network architectures (Bronstein et al., 2021), or is it better to learn them implicitly from data?

A common intuition is that strong inductive biases bring the biggest benefits when little training data is available, and that symmetry properties can just as well be learned from data given sufficient samples and compute. Recently, high-profile models of protein folding (Abramson et al., 2024) and conformer generation (Wang et al., 2023) have received considerable attention for their choice of non-equivariant architectures for geometric problems (Oxford Protein Informatics Group, 2024; Joshi, 2025; Twitter / X users, 2024).

At the same time, there is reason to expect that equivariance is still beneficial in the large-data limit. Learning means successively narrowing down a hypothesis class based on evidence. From this perspective one can explain (Bahri et al., 2024) the empirical observation that test losses often scale as a power law with the training compute (Kaplan et al., 2020; Hoffmann et al., 2022). Whereas non-equivariant methods start from the space of virtually all functions, equivariant models start from the subspace of all functions that abide by the symmetries of the problem. The learning process may benefit from that by focusing solely on further refining this smaller hypothesis class, narrowing down to the correct solution with fewer training steps.

---

*Work was completed while an employee at Qualcomm Technologies Netherlands B.V.

†Qualcomm AI Research is an initiative of Qualcomm Technologies, Inc. and/or its subsidiaries.

Until the theory of scaling laws is fully understood, the effects of equivariance on scaling is an empirical question, and in this work we study it empirically. We focus on a benchmark problem of modeling the physical interactions between rigid three-dimensional objects described by meshes. This task is known to be challenging (Allen et al., 2023). It is manifestly equivariant under E(3), the symmetry group of rotations, translations, and reflections. We compare a standard transformer architecture (Vaswani et al., 2017) to an E(3)-equivariant transformer (Brehmer et al., 2023).

In this setup we ask three questions:

1. *How do equivariant and non-equivariant models scale as a function of the available data?* Does data augmentation affect this?

2. *How do equivariant and non-equivariant models scale as a function of training compute?* Does this scaling follow power laws? Are their coefficients affected by equivariance?

3. *Given a compute budget, how should one allocate it to the model size and the number of training iterations?* Is this trade-off different for equivariant and non-equivariant models?

In our attempt to answer these questions, we train equivariant and non-equivariant models for different training compute budgets, trade-offs between model size and training steps, and dataset sizes. We then analyze these results both qualitatively as well as quantitatively by fitting empirical scaling laws.

Our experiments provide evidence for three conclusions. As expected, equivariance improves *data* efficiency. However, data augmentation largely closes this gap. Second, equivariant transformers are also more *compute-* efficient, and this advantage persists across all compute budgets studied. Both model classes exhibit power-law scaling behavior. Finally, the optimal allocation of a training compute budget to model size and training steps differs between equivariant and non-equivariant models. Overall, our findings hint that strong inductive biases may not only yield benefits in the low-data regime, but can also be beneficial with large datasets and large compute budgets.

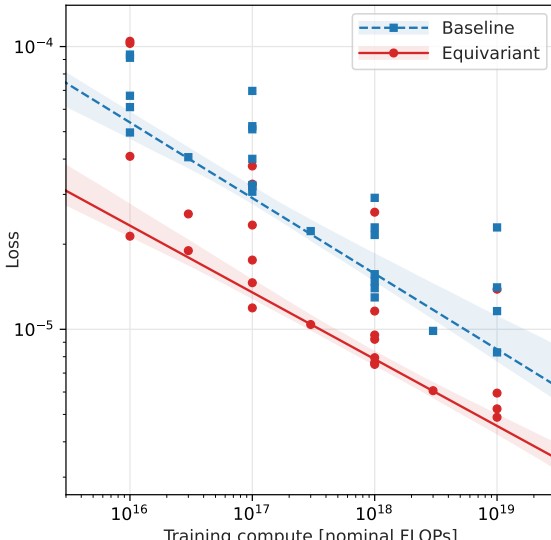

Figure 1: **Scaling with compute**. The dots show the training compute budget and test loss in our experiments, the lines indicate the compute-optimal performance according to the scaling laws we find, the error bands estimate the uncertainty on the power-law coefficients. The test losses of both non-equivariant (━━) and equivariant (━━) transformers scale as a power law with compute, and the equivariant model outperforms the non-equivariant model by a similar factor at all tested compute budgets.

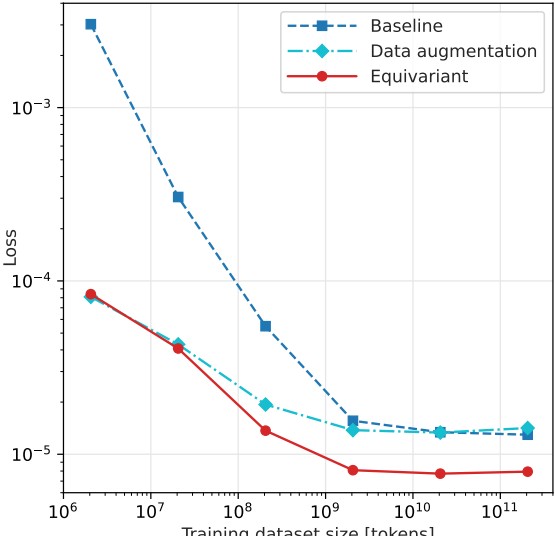

Figure 2: **Scaling with training data**. We show the performance of the non-equivariant transformer (━━), non-equivariant transformer trained with data augmentation (━·━), and equivariant transformer (━━) as a function of the number of unique tokens in the training dataset. All experiments use the same training compute budget, which means that the number of epochs reduces from left to right. Equivariance improves data efficiency compared to the baseline, but data augmentation can close this gap.

## 2 Background and related work

**Neural scaling laws** The scaling of neural network performance as a function of model size or training steps has been studied extensively (Ahmad & Tesauro, 1988; Hestness et al., 2017; Rosenfeld et al., 2020; Henighan et al., 2020). Kaplan et al. (2020) first observed that the test loss of autoregressive language models follows a power law over many orders of magnitude. Hoffmann et al. (2022) improved the methodology further and found the "Chinchilla" scaling laws, which still serve as a reference point for many language models. In our quantitative analysis of compute scaling, we largely follow their approach.

Several works have extended scaling laws from model size and training steps to other dimensions: Muennighoff et al. (2023) studied the effect of the training dataset size, which we also discuss, Alabdulmohsin et al. (2023) analyzed scaling of different architecture hyperparameters separately, and Jones (2021) investigated the scaling with problem complexity.

**Scaling laws and inductive biases** There has been comparatively little research into the relation between inductive biases and scaling behavior, perhaps because the transformer architecture (Vaswani et al., 2017) is so established in language modeling. Tay et al. (2023) compared the scaling behavior of different architectures. Recently, Qiu et al. (2024) investigated how structured linear transformations in transformers affect scaling laws. The authors conclude that imposing structure in them can improve the scaling behavior. Our work differs from both of these studies through its focus on symmetric problems and equivariant architectures.

**Geometric deep learning** Geometric deep learning (Bronstein et al., 2021) is a paradigm for machine learning in which network architectures are designed to reflect geometric properties of the problem. One of its core ideas is that of equivariance to symmetry groups (Amari, 1978; Wood & Shawe-Taylor, 1996; Makadia et al., 2007; Cohen & Welling, 2016): roughly, a network $f$ is said to be equivariant to a symmetry group $G$ if $f(g \cdot x) = g \cdot f(x)$ for all elements $g \in G$ and all inputs $x$, where $\cdot$ is the group action. This means that when you transform the inputs into an equivariant network, its outputs transform consistently. An equivariant network thus does not have to learn the symmetry structure from data, like a non-equivariant network does.

Equivariance has been found to improve performance, data efficiency, and robustness to out-of-domain generalization in fields as diverse as quantum mechanics and quantum field theory (Pfau et al., 2020; Hermann et al., 2020; Boyda et al., 2021; Gerdes et al., 2023), molecular force fields (Batatia et al., 2022; Batzner et al., 2022; Liao & Smidt, 2023; Musaelian et al., 2023; Batatia et al., 2023), generative models of molecules (Zeni et al., 2025; Igashov et al., 2024), particle physics (Bogatskiy et al., 2022; Gong et al., 2022; Spinner et al., 2024), biological and medical imaging (Veeling et al., 2018; Bekkers et al., 2018; Winkels & Cohen, 2018; Winkens et al., 2018; Mohamed et al., 2020; de Ruijter & Cesa, 2024; Suk et al., 2024), wireless communication (Hehn et al., 2025), and robotics (Wang et al., 2022a;b;c; Brehmer et al., 2024). The potential of equivariance to improve generalization has also been shown theoretically (Sokolic et al., 2017; Lyle et al., 2020; Elesedy & Zaidi, 2021; Sannai et al., 2021; Behboodi et al., 2022; Petrache & Trivedi, 2024).

At the same time, equivariant architectures are often more complex than non-equivariant architectures. Some researchers believe that equivariant architectures are more difficult to scale up, but to the best of our knowledge there has been little systematic study into this. However, recent impactful works on protein folding (Abramson et al., 2024) and conformer generation (Wang et al., 2023) found that equivariant architectures did not offer any benefits and opted for non-equivariant models and data augmentation instead.

**E(3) equivariance** One symmetry that is important in many scientific and industrial applications is the group E(3) of isometries of Euclidean space. It consists of translations, rotations, and reflections. This group is the focus of our investigation.

As an E(3)-equivariant architecture, we use the Geometric Algebra Transformer (GATr) (Brehmer et al., 2023). It has two defining features. First, GATr uses multivectors from projective geometric algebra as representations, in addition to the usual unstructured representations. These multivectors are 16-dimensional objects that can represent various geometric primitives, including absolute positions in space, directions, as well as translations and rotations. Geometric algebra representations power a number of recent architectures (Brandstetter et al., 2022; Ruhe et al., 2023b;a; Brehmer et al., 2023; de Haan et al., 2024; Spinner et al., 2024; Zhdanov

et al., 2024; Liu et al., 2024a;b). Second, GATr is a transformer. It processes inputs in the form of a set of tokens. Pairwise interactions are not computed through local message passing, as in many other equivariant architectures, but through an equivariant dot-product attention mechanism that is compatible with efficient implementations like FlashAttention (Dao et al., 2022).We choose GATr as the equivariant model for our scaling investigation because of this similarity to the standard transformer.

On a side note, the second symmetry group that is relevant to the problem we study is that of permutations of the inputs. Both standard transformers (Vaswani et al., 2017) and GATr (Brehmer et al., 2023) are equivariant to it.

## 3 Problem setup

### 3.1 Benchmark problem

**Desiderata**  A benchmark task for this empirical scaling study should be characterized by a low floor and a high ceiling: a small model trained on few samples should perform poorly, while a large model trained on many samples should score orders of magnitude better. To study data scaling, we need a large number of training samples. To study equivariance, we look for a geometric problem in which the symmetries and representations are known and exact.

**Rigid-body modeling problem**  We choose a rigid-body modeling problem as our benchmark. Three-dimensional meshes are initialized at some position, orientation, and velocity; they then interact with each other under gravity and collision forces. Concretely, the inputs to the network consist of a set of triangular meshes for two time points $t_0$ and $t_1 = t_0 + \Delta t$, and the task is to predict all mesh vertices at time $t_2 = t_1 + \Delta t$. As a loss function and evaluation metric, we use the mean squared error of the predicted mesh vertex positions.

While this may sound like a problem straight from an undergraduate physics textbook, it serves as a microcosm of phenomena that appear throughout science and engineering: chaotic behavior in dynamical systems, extended 3D objects parameterized as meshes, and the symmetries of space and time. It also satisfies all desiderata for our study. Rigid-body interactions are known to be challenging to model: collisions are difficult to detect, since they do not usually occur at or near vertices; the forces acting during a collision are nearly discontinuous (Bauza & Rodriguez, 2017; Pfrommer et al., 2021; Allen et al., 2023). Synthetic data can be generated cheaply with physics simulators. Finally, the physics of the process is clearly equivariant under E(3), provided that the direction of gravity is treated as a feature and rotated along with the scene.

**Dataset**  We construct a dataset of rigid-body interactions following a proposal by Allen et al. (2023). We use the Kubric simulator (Greff et al., 2022), which is based on the PyBullet physics engine (Coumans & Bai, 2016–2024). We recreate the MOVi-B dataset used by Allen et al. (2023) as best as we can, using parameters from their paper and private communication; see Appendix A for details. Our dataset consists of $4 \cdot 10^5$ trajectories, each consisting of 96 time steps. Each trajectory includes between 3 and 10 objects, each consisting of between 98 and 2160 mesh faces. The average number of total mesh faces in a scene is 5470.

### 3.2 Models

In selecting architectures, our main objective is not to achieve state-of-the-art results on the particular rigid-body benchmark problem we chose. That would lead us to highly problem-specific architectures (Allen et al., 2023; Rubanova et al., 2024). Instead, we aim for general-purpose architectures that are applicable to broad classes of problems.

**Baseline architecture**  The transformer architecture (Vaswani et al., 2017) has become the de-facto standard across a wide range of machine learning tasks. It is versatile with respect to the input data, propagates gradients effectively, and scales well to large model sizes and input tokens. Most scaling studies have focused on transformers as well. We therefore use a standard pre-LN (Baevski & Auli, 2019) transformer with multi-query attention (Shazeer, 2019) as our non-equivariant architecture.

We represent each mesh face as a token and the positions and velocities of vertices with random Fourier features (Tancik et al., 2020), which improved performance in initial tests.

Even this baseline architecture is hardly "free from inductive biases." Because the tokens form not a sequence, but an unordered set, we do not use positional encoding. Therefore, the model is equivariant with respect to one of the symmetries of our problem: that of permutations of the input tokens. In this respect, there is no difference between the two architectures, and we do not compare to any models that are not permutation-equivariant.

**Equivariant architecture** For the E(3)-equivariant architecture, we again look for broad applicability (at least within the class of E(3)-symmetric problems). In addition, we would like the architecture to be as structurally similar to the transformer, to isolate the effects of equivariance on scaling as well as possible. We therefore opt for the (to the best of our knowledge) only E(3)-equivariant architecture that is based on dot-product attention with unlimited receptive fields, and which also otherwise follows the transformer blueprint closely: the Geometric Algebra Transformer (GATr) (Brehmer et al., 2023).

Again, we represent each mesh face as a token. GATr uses geometric algebra representations in addition to the usual scalar channels, and we can represent the geometric properties of a mesh face in these geometric representations. We describe this embedding in more detail in Appendix B.

As an aside, while we focus on the E(3) equivariance that the problem has when the direction of gravity is included as an input, one could alternatively treat the problem as E(2)-symmetric in the plane orthogonal to the direction of gravity. We do not focus on that approach here, as the larger symmetry group E(3) is relevant in many problems and can easily be broken down to subgroups as needed.

**Hierarchical attention** While we focus on general-purpose architectures, we find that both models benefit from two minor modifications to the transformer blueprint. First, we use a novel *hierarchical attention* mechanism, in which multiple attention heads use different attention masks: half of the heads are restricted to attend only to mesh faces in the same object, while the other half attends to all tokens (mesh faces). This allows us to embed awareness of the mesh structure into the transformer architecture, while preserving the efficiency of dot-product attention.

**Enforcing object rigidity** Second, we enforce *object coherence and rigidity* when computing the outputs. Either transformer model first outputs a translation vector and a rotation quaternion for each mesh face. These are averaged over each object, resulting in a translation vector and a rotation for each rigid object. These E(3) operations are then applied to the input meshes. In this way, the networks by design translate and rotate rigid objects consistently. We describe this procedure in more detail in Appendix B. In preliminary experiments, enforcing object rigidity in this way improved performance substantially compared to directly predicting the positions or velocities of mesh vertices. We also experimented with outputting and exponentiating elements of the Lie algebra for each object, but found that that worked marginally worse.

**Hyperparameters** We tune the hyperparameters of both models by manually varying the depth, width, and number of heads at low and intermediate compute budgets. For both the baseline and equivariant transformer, we define a one-parameter family of hyperparameters, fixing the relation between the number of layers, attention heads, and channels to be linear. Our architectures are shown in Tbl. 1. Notably, we find that the equivariant transformer benefits from a more narrow architecture, which may be evidence of the expressivity of its multivector channels.

Table 1: Architecture hyperparameters as a function of a model size parameter $n$. The equivariant architecture is less wide, but part of their channels are 16-dimensional multivector (MV) channels, which can express a variety of geometric primitives (Brandstetter et al., 2022; Ruhe et al., 2023b; Brehmer et al., 2023; Ruhe et al., 2023a; de Haan et al., 2024).

| Hyperparameter | Baseline | Equiv. |
|---|---|---|
| Attention blocks | $2n$ | $2n$ |
| Scalar channels | $64n$ | $4n$ |
| MV channels | – | $n$ |
| Attention heads | $2n$ | $2n$ |
| Scalars per key, query, value | 64 | 8 |
| MV per key, query, value | – | 2 |
| Hidden scalar channels in MLP | $128n$ | $8n$ |
| Hidden MV channels in MLP | – | $2n$ |

**Optimization**  We train all models with the Adam optimizer (Kingma, 2015), annealing the learning rate over the course of training from an initial value of $5 \cdot 10^{-4}$ on a cosine schedule. For experiments with small FLOP budgets of less than $10^{18}$ nominal FLOPs, we find that this learning rate can be too small. This is in line with other works that find larger learning rates beneficial for smaller compute budgets (e. g. Dubey et al., 2024). We therefore repeat these experiments with a higher learning rate of $10^{-3}$ or $2 \cdot 10^{-3}$, depending on the compute budget, and report the better result. For simplicity, we use the same batch size of 64 samples (or on average $3.5 \cdot 10^5$ tokens) for all experiments, even though this does not maximize GPU utilization and thus FLOP throughput. Early stopping is used in all experiments.

### 3.3 Scaling-law analysis

**Experiments**  We perform two series of experiments. First, we study the scaling with compute, in the (practically) infinite-data setting. We vary a training compute budget over three orders of magnitude, between $10^{16}$ and $10^{19}$ FLOPs. For each FLOP budget, for both the baseline and the equivariant transformer, we perform multiple experiments: each with a different trade-off between model size $N$ and training length $D$. This requires understanding the relation between $N$, $D$, and the total training FLOPs; we discuss that later in this section.

Second, we study the scaling with training data, fixing the training compute budget, the model size, and the number of training tokens. For both models we choose settings that performed compute-optimally in the first series of experiments for a compute budget of $10^{18}$ nominal FLOPs. The number of unique samples in the dataset is varied over five orders of magnitude, from $2 \cdot 10^6$ tokens to $2 \cdot 10^{11}$. The lower end of this scan corresponds to training for $6 \cdot 10^5$ epochs, while every sample is seen only once on the upper end of this scan. For each of these settings, we train three models: a baseline transformer, an equivariant transformer, and a baseline transformer trained with data augmentation. In the latter case, symmetry transformations are applied to the samples, independently for each epoch. Because the direction of gravity is not an explicit input to the transformer, the symmetry group consists of two-dimensional translations as well as rotations around the direction of gravity.

**Counting FLOPs**  Setting up our experiments (see above) and analyzing the scaling with compute both require knowing the relation of the total number of training FLOPs $C(N, D)$ and the model size $N$ as well as training tokens $D$. This relation is different for the baseline and equivariant transformer.

Following Kaplan et al. (2020) and Hoffmann et al. (2022), we perform this FLOP counting in the limit where the number of model parameters is much larger than the sequence length, which in turn is much larger than 1. The training compute is then dominated by the linear layers. For both of our models, we find

$$C(N, D) \approx \xi N D \,, \tag{1}$$

where $\xi$ is an architecture-dependent constant.

For the baseline transformer, famously $\xi = 6$ (Kaplan et al., 2020). For the equivariant transformer, the value of $\xi$ depends on the ratio of scalar and multivector channels: a model with only scalar channels would also have $\xi = 6$, while a pure-multivector model would have more weight sharing and thus a higher FLOPs-per-parameter ratio $\xi = 6 \cdot 16^2/9 \approx 171$. For the hyperparameters we use during our scaling study, we find $\xi \approx 61.2$.

Note that these *nominal FLOPs* do not necessarily correspond to the actual compute required to train the model. For one, the assumed hierarchy between the model parameters and the sequence length is not always satisfied. Second, our implementations of the models may not be able to fully utilize the GPUs. We observe this in particular for small models and for the implementation of the equivariant transformer, which involves many smaller operations and faces CPU bottlenecks. Additional overhead comes from inter-GPU communication, data loading, logging, checkpoint saving, validating, and so on. In our experiments, two models with the same nominal FLOP count would differ by as much as an order of magnitude in real training duration.

Table 2: Scaling-law coefficients. In addition to the central values, we show the 95% confidence intervals from a nonparametric bootstrap.

| Scaling law | Param. | Baseline | | | Equivariant | | |
|---|---|---|---|---|---|---|---|
| | | Central | Lower | Upper | Central | Lower | Upper |
| Eq. (2): $\hat{L}(N, D) = A/N^{\alpha} + B/D^{\beta}$ | $A$ | **1.27** | 0.484 | 5.07 | **0.000282** | 0.000162 | 0.000607 |
| | $B$ | **0.202** | 0.0108 | 0.361 | **469** | 159 | 592 |
| | $\alpha$ | **0.909** | 0.832 | 1.03 | **0.348** | 0.293 | 0.417 |
| | $\beta$ | **0.379** | 0.256 | 0.404 | **0.734** | 0.689 | 0.747 |
| Eq. (4): $N^*(C) \propto C^a$ | $a$ | **0.294** | 0.215 | 0.307 | **0.678** | 0.619 | 0.711 |
| | $b$ | **0.706** | 0.693 | 0.785 | **0.322** | 0.289 | 0.381 |
| Eq. (5): $L^*(C) = F/C^{\gamma}$ | $F$ | **1.03** | 0.124 | 1.89 | **0.14** | 0.0524 | 0.517 |
| | $\gamma$ | **0.268** | 0.213 | 0.284 | **0.236** | 0.212 | 0.267 |

So why do we still analyze models in terms of the nominal FLOPs? While they are an imperfect measure, they do not depend on the implementation and hardware environment, and we believe they are still the best predictor of the theoretically achievable compute cost after sufficient optimization and at scale.

**Scaling-law ansatz**  We model the scaling with compute quantitatively by fitting a scaling law to all of our experiments. Following Kaplan et al. (2020), we model the test loss $L$ as a power law in the model parameters $N$ and the training duration $D$, measured in tokens:

$$\hat{L}(N, D) = \frac{A}{N^{\alpha}} + \frac{B}{D^{\beta}} + E \,. \tag{2}$$

Here $A, B, E, \alpha, \beta$ are fit parameters.

The parameter $E$ represents the irreducible loss that even a perfect model cannot eliminate. Unlike in language or image modeling tasks, there is no clear reason to expect such an irreducible error of practically relevant size for the deterministic physics task we use as a benchmark. We treat the choice of whether to include $E$ as a fit parameter or fix it to zero as a hyperparameter and choose it through cross validation, as we will describe below.

For the scaling with the size of the training dataset, we do not find a scaling law that convincingly describes our experiments. Our attempts at fitting Muennighoff et al.'s data-constrained scaling law (2023) to our data did not result in a good agreement. We therefore refrain from discussing the functional form for this direction of scaling, and will focus on scaling with compute for the remainder of this section.

**Scaling-law fit**  Following Hoffmann et al. (2022), we fit the scaling-law parameters $(A, B, E, \alpha, \beta)$ separately for each architecture by minimizing the Huber loss (Huber, 1992) between the predicted and observed log loss values,

$$\sum_{\text{experiments } i} \text{Huber}_{\delta}\Big(\log \hat{L}(N_i, D_i) - \log L_i\Big) \,. \tag{3}$$

Here $\delta$ is a hyperparameter, we choose it based on cross-validation, as we describe in a bit. We minimize this loss with the L-BFGS optimizer (Liu & Nocedal, 1989), starting multiple fits from a grid of initializations to avoid getting stuck in local minima.

**Scaling-law hyperparameters**  The scaling-law fit depends on two hyperparameters: whether we include the offset $E$ as a fit parameter and the value of $\delta$. We determine both through leave-one-out cross-validation, performing scaling-law fits on all but one experiment and evaluating the error $|\log \hat{L}(N_i, D_i) - \log L_i|$ on the left-out experiment. In this way, we choose fixing $E = 0$ and $\delta = 0.001$, though the qualitative fit results are not sensitive to these choices.

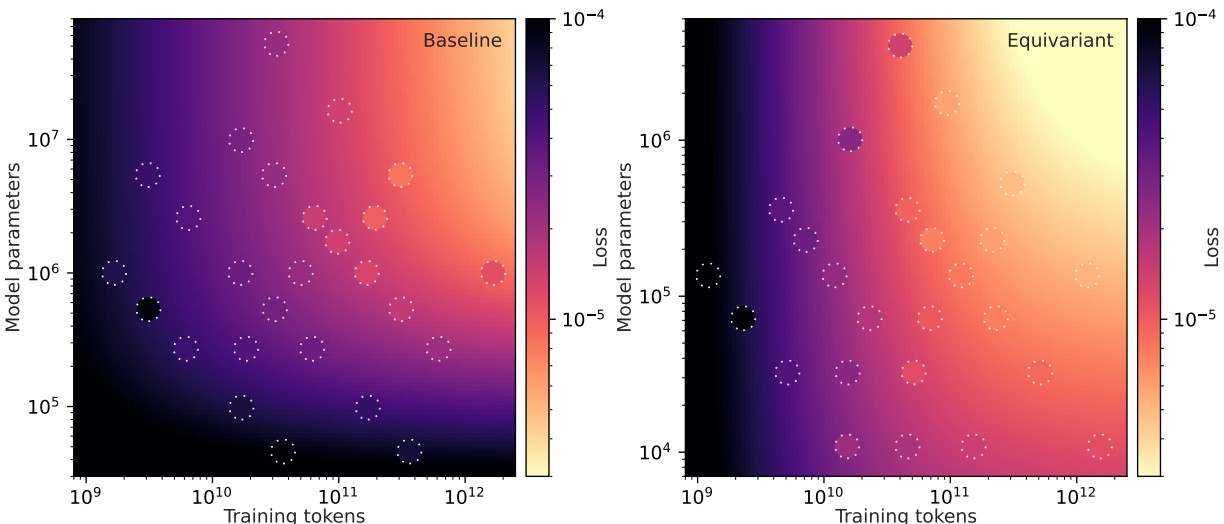

Figure 3: **Test loss** (dotted circles) **and scaling-law predictions** (background color) **as a function of model size and training tokens**. Left: non-equivariant transformer. Right: equivariant transformer. In both cases, we observe good agreement of model performance and scaling-law fit.

**Compute-optimal performance**   From a scaling law as in Eq. (2) and a FLOP function as in Eq. (1), we can derive the compute-optimal model size $N^*(C)$ and the compute-optimal training duration $D^*(C)$ as a function of the FLOP budget $C$ as

$$N^*(C) = \frac{G}{\xi^a}C^a \quad \text{and} \quad D^*(C) = \frac{1}{G\,\xi^b}C^b\,, \tag{4}$$

where $G = (\frac{\alpha A}{\beta B})^{1/(\alpha+\beta)}$, $a = \beta/(\alpha + \beta)$, and $b = \alpha/(\alpha + \beta)$ (Hoffmann et al., 2022).

The optimal loss achievable for a given FLOP budget is then

$$L^*(C) = \hat{L}(N^*(C), D^*(C)) = E + \frac{F}{C^\gamma} \tag{5}$$

with $F = AG^{-\alpha}\xi^\gamma + BG^\beta\xi^\gamma$ and $\gamma = \frac{\alpha\beta}{\alpha+\beta}$.

**Uncertainties**   No realistic scaling study directly measures the *optimal* model performance as a function of some parameters. Reasons for sub-optimality include the choice of hyperparameters, stochasticity in initialization and training, choosing a scaling-law ansatz that does not include the true functional form, and finite sampling of the space of model capacities and training tokens. We estimate the effect of the latter with a nonparametric bootstrap, similar to Hoffmann et al. (2022). From $10^4$ bootstraps, we construct 95 % confidence intervals on the scaling law coefficients as well as on any derived predictions, using the empirical (or basic) bootstrap method.

## 4   Results

### 4.1   Scaling with compute

We first focus on the limit of (essentially) infinite training data and study the model performance as a function of model size $N$ and training tokens $D$.

**Scaling laws**   We fit the scaling law of Eq. (2) with $E = 0$ to these experiments. For the baseline transformer, we find coefficients

$$\hat{L}_{\text{baseline}}(N, D) = \frac{1.27}{N^{0.909}} + \frac{0.202}{D^{0.379}}\,. \tag{6}$$

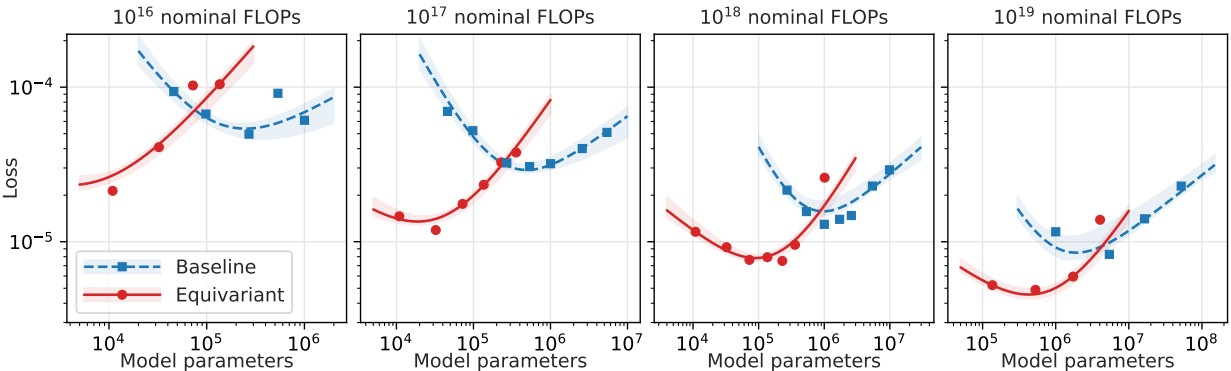

Figure 4: **Model performance at different training compute budgets** (panels) as a function of the model size. We show our experiments (dots) and the predictions of our scaling-law fit (lines). The scaling-law fit describes the measurements well.

The equivariant model yields

$$\hat{L}_{\text{equivariant}}(N, D) = \frac{2.82 \cdot 10^{-4}}{N^{0.348}} + \frac{469}{D^{0.734}} . \tag{7}$$

Confidence intervals are provided in Tbl. 2.

These two models scale quite differently with model size and training length, which has implications for the optimal allocation of a compute budget. We will discuss this later.

**Fit quality**    First, we show how well these fitted scaling laws agree with the data in Figs. 3 and 4. Comparing the observed values of the test loss to the predictions from the scaling laws, we overall find good agreement. There are no glaring deviations, although the power law underestimates the loss for the largest equivariant models and for one baseline outlier. Most measurements fall within the uncertainty bands, but less than the 95% one would expect if the bootstrap would cover all relevant sources of error. This is evidence that the ansatz of Eq. (2) does not describe the data perfectly.

**Scaling with compute**    Next, we analyze the model performance and its scaling with compute. From the training laws in Eqs. (6) and (7), we compute best achievable test loss $L^*$ as a function of the training compute budget $C$, as given by Eq. (5). We find

$$L^*_{\text{baseline}}(C) = \frac{1.03}{C^{0.268}} \quad \text{and} \quad L^*_{\text{equivariant}}(C) = \frac{0.14}{C^{0.236}} , \tag{8}$$

and the exponents are compatible with each other within the confidence intervals shown in Tbl. 2. We visualize the empirical compute–loss measurements and the derived optimal compute–loss relationship in Fig. 1.

For any given compute budget, the equivariant transformer significantly outperforms the baseline. Over the range of compute budgets we tested, the equivariant model achieves a loss that is lower by approximately a factor of 2.

**Optimal allocation of compute**    From the scaling laws we can also derive the optimal allocation of a given computational budget to the parameter count and training duration, see Eq. (4). We show our results for both models in Fig. 5.

We find that a compute-optimal equivariant transformer has less parameters than a compute-optimal baseline transformer. This is expected because the equivariant transformer performs more compute per parameter.

Perhaps more surprising is that the optimal trade-off depends on the compute in a different way for the two models. We find that for a regular transformer, one should scale training tokens more steeply than model

size. For the equivariant model, we find the opposite trend: one should put additional compute more in the model size than the training tokens. The compute-optimal model sizes thus become more similar for larger compute budgets.

## 4.2 Scaling with data

Next, we turn to the scaling with training data for a fixed training compute budget. In Fig. 2 we show the test loss as a function of the number of unique training tokens. We compare baseline and equivariant transformers, each using a compute-optimal model size and training tokens for a training compute budget of $10^{18}$ nominal FLOPs.

The right end of these curves corresponds to the infinite-data, single-epoch limit considered in the previous section. Here we again see that the equivariant transformer outperforms the baseline model when compared at the same training compute budget. Moving to smaller training sets, this gap widens substantially, confirming the expectation that equivariance improves data efficiency.

In Fig. 2 we also show results for a baseline transformer model trained with data augmentation. As expected, data augmentation does not make a differ-

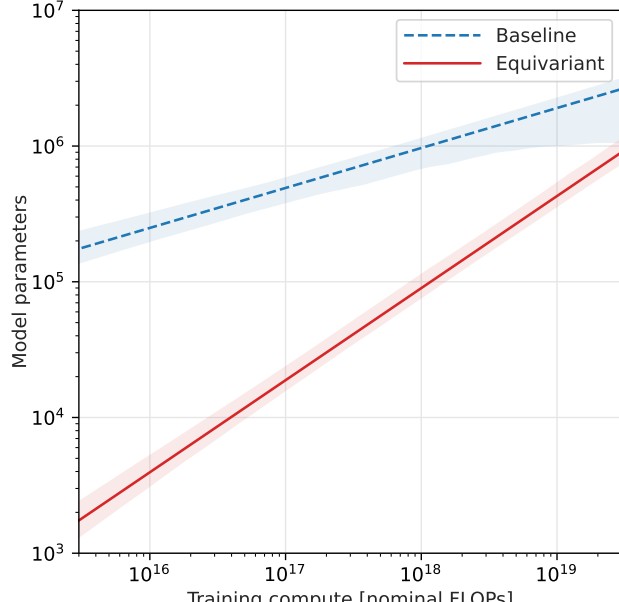

Figure 5: **Optimal parameter allocation**. We show the compute-optimal model size as a function of the training compute budget for the equivariant transformer (———) and the non-equivariant transformer (– –). The equivariant architecture requires smaller models to achieve a compute-optimal performance, but this gap closes for bigger compute budgets.

ence when training for a single epoch on a large dataset, which is the setting towards the right side of Fig. 2. However, data augmentation drastically improves the performance in the small-data regime: when training for thousands of epochs on a small dataset, data augmentation makes a baseline transformer as data-efficient as an equivariant model. This is the setting we see towards the left of Fig. 2.

## 5 Discussion

Our empirical results provide evidence for the following three conclusions.

**1. Equivariant transformers are more data-efficient, but data augmentation largely closes this gap.** The first (and expected) benefit for the equivariant architecture is that it performs better than a non-equivariant architecture when only little training data is available, as we show in Fig. 2.

However, we find a non-equivariant model trained with data augmentation performs just as well as the equivariant architecture, at least when the number of epochs (i.e. repeated uses of the same training sample) is sufficiently large. Our findings suggest that non-equivariant architectures are indeed able to learn symmetries from data if they are shown sufficiently many symmetry-transformed examples.

**2. The scaling with compute follows power laws, and equivariant models outperform non-equivariant ones at each tested compute budget.** Both for non-equivariant and equivariant models, the test loss is well described by the power-law ansatz of Eq. (1), with parameters given in Tbl. 2. The best achievable model performance for a given training compute budget therefore also scales as a power law, as given in Eq. (8). We find consistent exponents for the two models, but a substantially smaller prefactor for the equivariant architecture.

This shows a second (and perhaps less expected) benefit for the equivariant architecture: for any fixed compute budget, even in the infinite-data limit, it clearly outperforms the baseline method. As we show in Fig. 1, this benefit is approximately constant over the range of compute budgets we study.

Under the assumption that the implementations of equivariant and baseline architectures are similarly efficient and one can achieve the same FLOP throughput, this implies that equivariant models can outperform the non-equivariant counterparts even in the large-data, large-compute regime. In practice, non-equivariant architectures may be easier to optimize for high FLOP throughput, in which case it remains to be seen which architecture is more efficient.

**3. Equivariant and non-equivariant models require different trade-offs between model size and training duration.** Our power laws indicate that the optimal allocation of a given compute budget onto the model size and training steps is different for equivariant and non-equivariant transformers, as shown in Fig. 5. For small compute budget, a compute-optimal equivariant transformer is significantly smaller than a compute-optimal baseline transformer. This gap becomes smaller for larger compute budgets.

We hypothesize three possible explanations for this observation. First, the baseline transformer, the more mature architecture, may have a better initialization scheme and thus require less training steps to reach a good performance. Second, the different trade-offs may be related due to the different choice of width and depth between the architectures. A third possible explanation is linked to the internals of the equivariant transformer architecture, which can express certain primitives particularly efficiently: the free movement and gravitational acceleration of rigid bodies can be represented with few multivector channels, thanks to the geometric product operation integrated into the architecture. This explains why the architecture can achieve a good performance with very few parameters. However, lowering the loss further requires precise collision detection and modeling. These need substantially more computational operations and a substantial amount of scalar channels, similar to the non-equivariant transformer. This offers a possible explanation for why at a larger compute budget, a model size closer to that of the baseline transformer is compute-optimal.

**Limitations and open questions** As much as we would like to, we cannot conclusively settle the question raised in the title of this paper. Our work is limited in several ways. First, we only analyzed a single benchmark problem and two model families. We chose a task with a common symmetry group and general-purpose architectures that are frequently applied to a wide range of problems. We believe it is important to study to what extent our findings generalize to other problems or to other architectures, for instance those based on message-passing over graphs. Moreover, on the problem we studied, we did not set a new state of the art: we deliberately focused on general-purpose models, which do not achieve the same level of performance as highly problem-specific architectures (Allen et al., 2023).

Another limitation of our work is that our analysis measures compute with an idealized FLOP counting procedure, as is common practice (Hoffmann et al., 2022). As we discussed in Sec. 3.3, this does not map one-to-one to real-world run time, at least not before further optimization of the implementation. In Appendix C we show the relation between wall time and nominal FLOPs in our experiments.

Finally, we are only able to study training compute budgets of up to $10^{19}$ FLOPs per model—this does not come close to the approximately $10^{25}$ FLOPs that the currently largest language models are trained for (Dubey et al., 2024). We did not see power-law scaling break down in the range we studied, but we cannot make claims about the extrapolation beyond it.

**Conclusions** We showed that it can be useful to study strong inductive biases like equivariance through the lens of *compute* efficiency, rather than only through the established perspective of data efficiency.

Keeping the limitations of our empirical study in mind, we believe that our findings provide evidence that symmetry-aware modeling can be a sensible choice even for large compute and data budgets. To materialize these benefits in practice, it may be both necessary and worthwhile to develop more efficient implementations of equivariant architectures—that is, improve their FLOP throughput.

The benefits and disadvantages of strong inductive biases at scale are important for problems spanning several fields of science and engineering. We hope that our study can encourage further investigations into this question.

**Acknowledgements**

We thank Gabriele Cesa for great discussions as well as useful comments on a paper draft and Kelsey Allen for helping us understand the dataset used in FIGNet.

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

## A  Dataset

We generate a benchmark dataset of rigid-body interactions with the Kubric simulator (Greff et al., 2022), which is based on the PyBullet physics engine (Coumans & Bai, 2016–2024). We follow the MOVi-B configuration as used by Allen et al. (2023): we generate trajectories trajectories of 96 frames at 48 frames per seconds. Our training set consists of $4 \cdot 10^5$ such trajectories, while we use 1000 trajectories each for the validation and test set. Each trajectory includes between 3 and 10 three-dimensional objects, each consisting of between 98 and 2160 mesh faces. In addition, there is one planar object for the ground floor, canonically placed at $z = 0$. The objects are accelerated by collision forces as well as gravity, which canonically points in negative $z$ direction.

Our data can be generated with the openly available repository at `https://github.com/google-research/kubric`. That requires modifying the code to save object meshes, positions, and orientations for each time step, rather than the vision data that Kubric stores by default, and running `python3 challenges/movi/movi_ab_worker.py -objects_set=kubasic -frame_rate=48`.

## B  Models

**Relation between equivariant and base model**  In many equivariant architectures such as steerable CNNs (Cohen & Welling, 2016), the equivariant linear maps form a subspace of the corresponding space of kernels of a non-equivariant base model. This is not the case for GATr.

However, this difference only holds at the level of linear maps. Nonlinearities generally differ between equivariant and non-equivariant architectures. Therefore, equivariant architectures cannot usually be easily identified with a subspace of the parameter space of a standard non-equivariant architecture.

**Input representations**  For both the baseline and equivariant transformer, we tokenize the problem by assigning one token to each mesh face. In addition to the vertex positions, we compute the central position of each mesh face, the relative vector from the center to each vertex, the surface normal on the mesh face, and the linearly interpolated velocity between $t_0$ and $t_1$ for each vertex and the center of each mesh face. Together these form the input features.

For the baseline transformer, these features are embedded using random Fourier features (Tancik et al., 2020) with 128 frequencies sampled from a Gaussian with standard deviation 0.1.

For the equivariant transformer, these features are embedded in the projective geometric algebra (PGA) described in Dorst (2020); Ruhe et al. (2023b); Brehmer et al. (2023). Specifically, vertex and center positions are represented as PGA trivectors, the relative vector from the center to each vertex as PGA vectors, the mesh face surfaces with the associated normals as PGA vectors, and all velocities as PGA bivectors.

**Enforcing object rigidity**  The transformer networks output eight features $h$ for each token (mesh face) that represent transformations like translations and rotations. The final predictions for future vertex positions $\hat{x}(t_2)$ are then computed by applying these transformations to the current vertex positions $x(t_1)$.

This is most easily expressed in projective geometric algebra, the representation naturally used by our equivariant transformer models. Here $h$ are the even-grade components of the output PGA multivectors. The predictions are computed as

$$h_{\text{agg}} = \text{mean}_{\text{objects}} h \,,$$
$$\hat{x}(t_2) = h_{\text{agg}} x(t_1) \tilde{h}_{\text{agg}} \,. \tag{9}$$

In the first line, the mesh-face-level predictions are averaged within each rigid object. In the second line, the previous position $x(t_1)$ is translated and rotated with the E(3) element represented by the network outputs; $\tilde{h}$ is the PGA reverse and $hx\tilde{h}$ (for properly normalized $h$) the sandwich product used to apply transformations to objects (Dorst, 2020; Ruhe et al., 2023b; Brehmer et al., 2023).

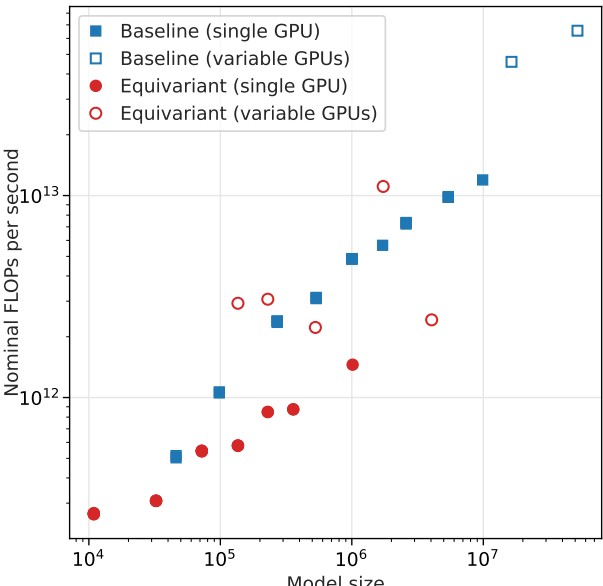

Figure 6: **Throughput of nominal FLOPs per training wall time**. The FLOPs we count do not directly correspond to training wall time: larger models and non-equivariant transformers lead to a better GPU utilization, increasing the FLOP throughput.

We also experimented with predicting all vertex positions directly, without enforcing object rigidity, as well as with parameterizing elements of the Lie algebra of E(3), which would then be exponentiated to construct transformations $h$. Both approaches performed worse in initial tests.

## C Scaling-law analysis

In Sec. 3.3 we argue that the nominal FLOPs we count are not fully indicative of real-world run time. We illustrate this in Fig. 6, where we show relation between these FLOPs and wall time in our experiments. The throughput of nominal FLOPs varies by two orders of magnitude. Larger models as well as non-equivariant transformers lead to a better GPU utilization and thus increase the FLOP throughput. We expect that the FLOP throughput could be improved by optimizing the batch size or (especially in the case of the equivariant transformer) the model implementation.

