# OpenReview forum: "Does equivariance matter at scale?"
_TMLR — Accepted by TMLR_

### Review · Reviewer_E6DY · 2025-04-25

**Summary Of Contributions:**

Motivated by the departure of recent protein-folding works from equivariant architectures, the authors evaluate the benefit of incorporating equivariance as data and compute are scaled up. To do so, they evaluate Geometric Algebra Transformer (GATr) and a non-equivariant baseline on a rigid-body modeling task. They find that 1) GATr is more data-efficient, but not by much after data augmentation; 2) GATr outperforms given a fixed compute budget; and 3) GATr's optimal model-size / training-time trade-off involves more compute than that of the non-equivariant baseline.

**Audience:**

Yes

**Claims And Evidence:**

Yes

**Requested Changes:**

The authors are encouraged to respond to Weaknesses 2 and 3 and the points in Misc.
## Misc
1. The authors write that "Early stopping is used in all experiments." but display only discrete FLOPs increments in Figure 1. The authors should also verify that they are using the true number of steps when computing power laws.
2. The authors write that the "equivariant transformer performs more compute per parameter." Could they quantify this?
3. What kind of augmentation is applied to the baseline model? Is any augmentation applied to GATr?
## Style and Typos
1. To improve presentation, the authors should consider compressing the first two figures just enough to move the final sentence of the first paragraph onto the front page.
2. The authors write that Abramson et al. (2024) and Wang et al. (2023) "have received considerable attention for their choice of non-equivariant architectures for geometric problems," but provide no citations for the referenced attention.
3. Citations to preprints that have since been published -- such as Alabdulmohsin et al. (2023), Allen et al. (2022), and Baevski et al. (2018) -- should be updated with the venue information.
4. There is also inconsistent formatting in venue names (such as "Nature communications" vs. "Nature Communications").
5. Some bibliography entries are missing information like "Training Compute-Optimal large language models" which has no listed venue.
6. Proper nouns are not consistently capitalized in the bibliography.
7. Only some bibliography middle initials are given dots.
8. unnique -> unique
9. British English is somewhat inconsistently used ("behaviour" but "center").
10. The axis label of Figure 4 ("Loss") is cut off.
11. The terms "data sets" and "datasets" should be made consistent.
12. "compute-loss" should likely be "compute--loss" (en-dash) as it seems that compute should not modify loss.
13. biases”. should be biases.”
14. Captions should go above the tables in TMLR style [3].
15. The `fill_between` in Figure 1 should be explained in the caption.

[3] https://github.com/JmlrOrg/tmlr-style-file/blob/7bf90efe3a0debbba703c05c43f3ff7e4d4a2992/main.tex#L195

**Strengths And Weaknesses:**

## Strengths
1. The paper is very well written. It is clear the authors put a lot of care into the text.
2. The motivating question -- of whether introducing equivariance inductive biases into model architectures remains relevant at scale -- is timely and of interest to the TMLR community.
3. The experimental design appears sound, and the claims fairly well qualified.
## Weaknesses
1. The questions asked in the paper ("Given large data sets and sufficient compute, is it beneficial to design neural architectures for the structure and symmetries of each problem? Or is it more efficient to learn them from data?") are very big, but the scope of the evaluation is fairly limited: it involves just one class of equivariance, one model, and one task. The authors acknowledge this, and seem to appropriately qualify their claims ("our empirical results provide evidence"), but it limits the potential impact of the work. It is not clear whether it should be expected that the findings might transfer to another setting, such as E(2)-equivariant vision transformers [1].
2. Is the direction of gravity augmented at test time? If not, it seems the optimal baseline would not need full E(3) equivariance, but only be "subequivariant" [2]. Rigid-body modeling without uniform gravity appears likely a more interesting setting for evaluating the effect of E(3) equivariance. The tasks of protein folding (Abramson et al., 2024) and conformer generation (Wang et al., 2023) would have been of even greater relevance.
3. The authors write that "Whereas non-equivariant methods start from the space of virtually all functions, equivariant models start from the subspace of all functions that abide by the symmetries of the problem," but it should be noted that it is not possible to represent GATr in terms of the baseline architecture, as GATr involves different, specialized operations. This contrasts with other works, such as Cohen & Welling (2016), where a model was made equivariant through a way that might also be learned.

[1] Xu, Renjun, et al. "$ E (2) $-Equivariant Vision Transformer." Uncertainty in Artificial Intelligence. PMLR, 2023.

[2] Han, Jiaqi, et al. "Learning physical dynamics with subequivariant graph neural networks." Advances in Neural Information Processing Systems 35 (2022): 26256-26268.

---

> ### Comment · Reviewer_E6DY · 2025-06-02
>
> The authors' rebuttal is appreciated. However, it would have been preferable had they responded directly to comments rather than aggregate points across reviewers, which makes cross-referencing more difficult. Outstanding points are reraised below.
> ### Weaknesses
> 2. Would an optimal baseline necessarily be E(3)-equivariant? If not, this should be made clear in the text and be discussed as a limitation of the chosen task setting, and the statement motivating the choice of GATr as "the only E(3)-equivariant architecture that closely follows the transformer blueprint" deemphasized.
> 3. The rebuttal justification is appreciated. Would the authors please also clarify this in the text?
> ### Misc
> 3. Why were only two-dimensional translation augmentations applied to the baseline model, rather than three-dimensional?
> ### Style and Typos
> 2. The review comment was written in response to the authors' specific claim that the highlighted works "have received considerable attention for their choice of non-equivariant architectures for geometric problems," not whether they received attention in general.

---

> > ### Author Response · Authors · 2025-06-03
> >
> > Thank you for your fast response!
> >
> > We appreciate the comment on the format of our rebuttal and will keep it in mind for next time. Thank you also for the comments and questions.
> >
> > > Would an optimal baseline necessarily be E(3)-equivariant?
> >
> > Good point. Indeed, E(2)-equivariant architectures also make sense on this problem (see also the discussion of 2D vs 3D data augmentation below). We added a paragraph to Sec. 3.2.
> >
> > > The rebuttal justification is appreciated. Would the authors please also clarify this in the text?
> >
> > We have added it to appendix B.
> >
> > > Why were only two-dimensional translation augmentations applied to the baseline model, rather than three-dimensional?
> >
> > The direction of gravity affects the movement of the objects. For the (un-augmented) transformer, this parameter is canonically chosen as the z direction, and we do not provide it as an input to the transformer.
> >
> > When we train the transformer with data augmentation, there are then two possible strategies: either one provides the direction of gravity as an input and perform three-dimensional data augmentations – or one sticks to two-dimensional data augmentation. Both are a priori sensible, and a comparison would be worthwhile. To keep computational expenses low, we simply opted for the second strategy to keep the setup as similar to the un-augmented transformer training.
> >
> > > The review comment was written in response to the authors' specific claim that the highlighted works "have received considerable attention for their choice of non-equivariant architectures for geometric problems," not whether they received attention in general.
> >
> > Thank you for clarifying. It is true that the citation count on Google Scholar. does not directly highlight the discussion of the equivariance question. However, it is one of the topics in the [blog post](https://www.blopig.com/blog/2024/08/architectural-highlights-of-alphafold3/) we cited. To quote from that blog post:
> >
> > "In AlphaFold2, the structure module takes great care to ensure that SE(3) (relative distance) is preserved. AlphaFold3 scraps this, instead simply linearly embedding the coordinates and passing them into the token embeddings to be processed like any other token feature. For a model to have learned any physics, and there are some indications that AF3 has, a model must be able to measure distance which means it must have learned some SE(3). To encourage the model to learn this, all structures are recentred and then randomly rotated and translated at every diffusion step. In this way the model must learn to use positional information which has been subject to arbitrary SE(3) transformations. Just because a model hasn’t been designed to be SE(3) equivariant, doesn’t mean the model hasn’t learned to be SE(3) equivariant."
> >
> > Do you have an idea for how to best reference informal community discussions? It is not really possible to cite panel discussions at workshops, coffee chats at conferences, and private discussions with members of the community which focused on this aspect. Perhaps it is instructive to point to dicussions in social media, such as [this discussion](https://x.com/tkipf/status/1730675507522679181)?

---

> > > ### Comment · Reviewer_E6DY · 2025-06-08
> > >
> > > The authors' further modifications are appreciated.
> > >
> > > ### Misc
> > > 3. It is clear why the authors chose not to perform three-dimensional rotational augmentation, but not why they didn't apply three-dimensional *translational* augmentation. Should an ideal network be equivariant to vertical / z-axis translation of the input? It seems this might impact evaluation of data efficiency.
> > > ### Style and Typos
> > > 2. Yes, it is valuable to point to relevant social media discussions, such as the one linked to. That, along with the blog post, are sufficient, though an archive [1] of the Tweet should likely be linked to in the entry given the volatile nature of the platform. Reference to private discussion should be avoided, as it does not offer the reader an opportunity to dig further.
> > >
> > > [1] https://ghostarchive.org/archive/GAH4t

---

> > > > ### Author Response · Authors · 2025-06-11
> > > >
> > > > Thanks for your engagement in this discussion!
> > > >
> > > > > It is clear why the authors chose not to perform three-dimensional rotational augmentation, but not why they didn't apply three-dimensional translational augmentation. Should an ideal network be equivariant to vertical / z-axis translation of the input?
> > > >
> > > > Good question. Translations along the z axis are not a symmetry of the test data, as all test examples have the ground floor (a plane object) at z = 0. We therefore expect that augmenting the training data with 3D translations might hurt rather than help the performance of the model.
> > > >
> > > > > Yes, it is valuable to point to relevant social media discussions.
> > > >
> > > > We have updated our manuscript again, citing the discussion on Twitter (thank you for the archival link) as well as another blog post that discusses the topic.

---

> ### Comment · Reviewer_E6DY · 2025-06-11
>
> The authors' responsiveness is appreciated.
> ### Misc
> 3. It is agreed that the presence of a ground plane complicates augmentation. However, the paper does not seem to include any mention of there being such. To improve the reproducibility of the work, the authors are encouraged to add this detail.

---

> > ### Author Response · Authors · 2025-06-13
> >
> > Thank you for the suggestion. We agree that it's better to be explicit rather than just relying on the reference for the dataset to explain this aspect and have added this to appendix A.

---

### Review · Reviewer_Mt2K · 2025-04-28

**Summary Of Contributions:**

The paper aims at understanding whether integrating equivariance in the design of neural networks can provide gains in training performance (for problems possessing some known intrisic symmetries). To answer this question, the authors take an empirical perspective through the lens of scaling laws.
The authors carefully chose a problem (predicting movements in three body physics) and two architectures (a plain transformer and an equivariant counterpart). The authors then fit sclaing laws for each architecture following previous approaches taken for language models. They find that the equivariant models outperform the plain models not only at small data scale but also at large scale of training data (approaching the regime of infinite data). They also investigate whether data augmentation can close the gap in performance between the plain and equivariant architecture. They find that in a small compute regime data augmentation closes the gap in performance.

**Audience:**

Yes

**Claims And Evidence:**

Yes

**Requested Changes:**

**Questions/Comments**:
- The conclusions of the authors appear to contradict previous papers (quoting the authors "works on protein folding (Abramson
et al., 2024) and conformer generation (Wang et al., 2023) found that equivariant architectures did not offer
any benefits and opted for non-equivariant models and data augmentation instead.") It would be great to explain where do these contradictions stem from.
- "Notably, we find that the equivariant transformer benefits from a more narrow architecture" page 5 and Table 1. How was that found or chosen?
- Can a plain transformer achieve the performance of an equivariant one with enough data augmentation? If yes, wouldn't the differences of compute efficiency exactly amount to the number of additional augmented training data needed for the plain transformer? Fig. 2 seems to only give partial answers. In Fig. 2, from what I understand in the caption and the manuscript,  at $10^{11}$ tokens no augmented data is used. Would it be possible to see how much data augmentation is needed to reach the accuracy of the equivariant model for the equivalent $n$ (i.e. equivalent size of model)?
- *Maybe* put into equation the setup of part 4.2 to iron out any misunderstanding.
- It would be good to detail the data augmentation mechanisms used for the plain transformer.
- Detail if possible the computation of the nominal cost of the equivariant transformer.

**Minor comments**:
- In page 3, define $t_1=t_0+\Delta t$, $t_2 = t_1 + \Delta t$ or at least define those in the appendix where they're used.
- page 17: rigidiy -> rigidity
- page 9: uniique -> unique
- Fig 4: Loss is cut on the y-axis
- modelling/modeling -> choose between british or american spellings throughout the paper
- Maybe add an image illustrating the task at hand (three body physics) for the reader to grasp the empirical matter.

**Strengths And Weaknesses:**

**Strengths**:
- The authors ask a clear scientific question and answer it as rigorously as they can.
- The introduction is very well written, points to relevant related work.
- Even though equivariance for deep learning is an old subject, the viewpoint taken by the authors is new and relevant.
- The empirical approach is well detailed (some questions though below). The authors strive to give a fair judgment of the question and avoid potential biases.
- The empirical findings are relevant for the community.

**Weaknesses**:
- The weaknesses are clearly spelled out by the authors themselves (which is much appreciated). Namely, they considered a single benchmark, they did not scale to a point where some scaling laws could break, the compute efficiency (both in terms of initialization, optimization scheme, gpu utilization) may not be fully explored.

---

### Review · Reviewer_xLdn · 2025-05-23

**Summary Of Contributions:**

This work presents an analysis of the scaling behavior of equivariant transformer models compared to non-equivariant counterparts at a many-body interactions learning task. As usual, the scaling behavior is analyzed based on multiple aspects: training data, model size and compute budget in this case.
As far as I know, this is the first such comparison between equivariant and non-equivariant models for many-body interactions, with the conclusions being that equivariant models are more efficient in the small data regime, but data augmentation helps the non-equivariant models mostly close this gap, and that they perform better given a limited computation budget in general. The paper provides some further insights in what to prioritize given a limited compute budget for both equivariant and non-equivariant models.

However, overall I am not quite sure how generally applicable these conclusions are given the limited nature of the study, both in terms of the experimental setup, and the types of models used.

**Audience:**

Yes

**Broader Impact Concerns:**

No broader impact concerns.

**Claims And Evidence:**

Yes

**Requested Changes:**

Asking the authors to expand the analysis across other problems or models would be unreasonable given the costs, so the only request I would make is to discuss some of the results in more detail.
1. No effort is made to explain why data augmentation helps in the small data regime, and the authors neglect to mention that the data-augmented model reverts to the same performance as the baseline non-equivariant model when more data is used. The authors need to at least mention this when reporting the results, and ideally discuss it further.
2. Furthermore, the authors should better justify the choice of the dataset, ideally by discuss how and why the properties of their chosen problem setting can transfer well to other many-body interaction or physical scenarios in general. The current discussion in the dataset section is unfortunately not sufficient.

**Strengths And Weaknesses:**

Strenghts:
- The writing in the paper is precise and easy to understand, and the authors do well to explain the details of their scaling law computation, dataset generation and the analysis of the different scaling scenarios.
- I also did not find any typos or writing mistakes, which shows that some care was put into the writing and proofing of this work.
- The conclusions drawn from the different scaling scenarios are are sound based on the presented results of the experiments.

Weaknesses:
- The paper seems to be a bit too concise, with the author's analysis and discussion being only surface level. I would have preferred to see some more detailed discussion of why we see the results that we see.
- The fact that this evaluation was performed on a very limited scenario and with only two transformer type models makes it difficult to trust that the scaling relations and conclusions will remain the same outside of this specific scenario. However, this is the nature of such scaling law papers given the large compute requirements for producing them, and as such not necessarily the fault of the authors, and the authors are also transparent about this in the discussion of the limitations.

---

### Author Response · Authors · 2025-06-01
**Response to the reviewers (1/2)**

We would like to thank all reviewers for their detailed and insightful comments.
We are glad that the reviewers appreciate that we "ask a clear scientific question and answer it as rigorously as [we] can" (reviewer **Mt2K**).
They also find the conclusions drawn from our scaling study "sound" (reviewer **xLdn**) and "relevant for the community" (reviewer **Mt2K**).
Finally, we are excited that they find the paper "very well written" (reviewer **E6DY**).

We also appreciate the constructive criticism the reviewers offered. In the following, we will address their major concerns and questions.

## Limited scope

Reviewers **xLdn**, **Mt2K**, and **E6DY** all point out that our study is limited in scope. In particular, we only experiment on a single problem and only with two architectures. This makes it difficult to judge whether our findings generalize.

We wholeheartedly agree: we would love to extend our study to more tasks, more architectures, and even larger scale.
Unfortunately, scaling studies require a substantial computational and dollar cost. We simply do not have the resources for these extensions.
We believe that it is more valuable for the community if we do one analysis well, covering multiple orders of magnitude in model parameters, training steps, and dataset sizes, than to perform many experiments at a reduced scale.
We were glad to read that the reviewers appreciate these reasons for the limitations of our study.

## GATr architecture

Reviewer **E6DY** mentions that the GATr architecture we use differs from many other equivariant architectures, in which the equivariant kernels form a subspace of the corresponding space of kernels in a base model (e.g. a CNN).

That is a good point, but this difference only holds at the level of linear maps. Nonlinearities generally differ between equivariant and non-equivariant architectures. Therefore, equivariant architectures cannot usually be easily identified with a subspace of the parameter space of a standard non-equivariant architecture.

These subtleties aside, we choose GATr because it is (to the best of our knowledge) the only E(3)-equivariant architecture that closely follows the transformer blueprint (i.e., dot-product attention).

Reviewer **E6DY** also asks about about the ratio of compute and parameters in GATr. We discuss this in Sec. 3.3 in the paragraph titled "Counting FLOPs". For the hyperparameters we use, during training, GATr uses 61.2 FLOPs per token and parameter, while a standard transformer uses 6 FLOPs per token and parameter.

## Correctness of the scaling study

Reviewer **E6DY** asks if our scaling study takes into account the effect of early stopping on the number of steps.

That is an excellent potential issue to check. We are happy to confirm that our analysis takes early stepping into account correctly.

Early stopping only had an effect in our data scaling study (Sec. 4.2), and here the step count does not enter.
When studying scaling with compute (Sec. 4.1), where the number of steps would matter for the analysis, the models keep improving until the end of training. This is expected as we are studying the (essentially) infinite-data limit.

## Can transformers learn symmetries?

Reviewer **Mt2K** asks if a plain transformer can achieve the performance of an equivariant one with enough data augmentation.

Our findings suggest that yes, they can; but as the reviewer pointed out, we only answer this question in a small-dataset setting.

In Fig. 2, we keep the number of training steps constant and vary the dataset size; as a consequence, the left part of the plot corresponds to many epochs and the right part of the plot to a single epoch.
We find that when we train for many epochs (left side of Fig. 2), the standard transformer trained with data augmentation catches up with the equivariant architecture.

Unfortunately, we do not have the computational resources to train on a large dataset for many epochs, so we cannot answer whether the standard transformer would catch up in the same way with the equivariant one there.

We have updated the discussion in the end of Sec. 4 and beginning of Sec. 5 to make this clearer.

(Continued in a second post)

---

> ### Author Response · Authors · 2025-06-01
> **Response to the reviewers (2/2)**
>
> (Continued from above)
>
> ## Relation to the conclusions of other works
>
> Reviewer **Mt2K** asks how our findings relate to recent works that opted for non-equivariant architectures.
> This is a good question. Since those papers did not publish quantitative comparisons of equivariant and non-equivariant architectures, we can only speculate.
> In line with our findings, there are several reasons to choose non-equivariant architectures for a problem with symmetries:
> - the good performance of non-equivariant architectures trained with data augmentation when compute is not the bottleneck (see above),
> - the relative ease of implementation of non-equivariant architectures, and
> - the higher FLOP throughput of established implementations of non-equivariant architectures.
>
> Note that in many problems, E(3)-equivariant architectures are still the state of the art, even when trained at scale. Consider e.g. interatomic potentials like the recent UMA (B. Wood et al, "UMA: A Family of Universal Models for Atoms").
>
> ## More details and expanded discussion
>
> Multiple reviewers suggest to expand the description of our methods and the discussion of our results. Among others:
> - reviewer **E6DY** asks for references for the attention that the AlphaFold3 and "Simplified Scalable Conformer Generation" papers received,
> - reviewer **xLdn** suggests justifying our choice of datasets,
> - reviewer **Mt2K** asks about the hyperparameter tuning,
> - reviewers **Mt2K** and **E6DY** ask about how our data augmentation works, and
> - reviewer **xLdn** asks for an expanded discussion of the data augmentation results.
>
> We thank the reviewers for these suggestions. They are incorporated in the updated manuscript. Concretely, we expand:
> - citations in the introduction (Sec. 1),
> - the paragraph on the choice of dataset (Sec. 3.1),
> - the discussion of how we tune hyperparameters (Sec. 3.2),
> - the description of our data augmentation (Sec. 3.2), and
> - the discussion of the data augmentation results (end of Sec. 4, beginning of Sec. 5)
>
> ## Minor comments
>
> We are grateful to the reviewers for pointing formatting issues, typos, outdated references, and other mistakes. They are fixed in the updated version of the manuscript.
>
> In closing, we want to thank all reviewers again for their feedback, which helped us to substantially improve the paper. We hope that we could address their questions and that the manuscript is now in a shape fit for publication.

---

### Author Response · Authors · 2025-07-10

We would like to thank all the reviewers again for their comments and discussion.

Are there any questions or concerns that we have not sufficiently addressed yet?

---

### Decision · Action_Editor_mjZJ · 2025-07-14

**Recommendation:** Accept as is

**Audience:**

Yes

**Audience Explanation:**

The question of training efficiency and the value of equivariant architectures is of relatively broad interest.

**Claims And Evidence:**

Yes

**Claims Explanation:**

The paper explores the value of explicitly equivariant architectures vs additional data and compute.  It's studies are conducted on a simplified setting to make them computationally tractable, but the claims are supported by clear evidence in that setting.